# Muscle Oxygen Desaturation and Re-Saturation Capacity Limits in Repeated Sprint Ability Performance in Women Soccer Players: A New Physiological Interpretation

**DOI:** 10.3390/ijerph18073484

**Published:** 2021-03-27

**Authors:** Aldo A. Vasquez-Bonilla, Alba Camacho-Cardeñosa, Rafael Timón, Ismael Martínez-Guardado, Marta Camacho-Cardeñosa, Guillermo Olcina

**Affiliations:** 1Faculty of Sports Sciences, University of Extremadura, 10003 Cáceres, Spain; rtimon@unex.es; 2Faculty of Languages and Education, University of Nebrija, 28015 Madrid, Spain; albacc@unex.es (A.C.-C.); mcamachocardenosa@unex.es (M.C.-C.); 3Faculty of Life and Natural Sciences, University of Nebrija, 28015 Madrid, Spain; imartinezgu@nebrija.es

**Keywords:** NIRS, muscle oxygen saturation, workload, physiological adaptations, fatigue and sport performance

## Abstract

Muscle oxygen consumption could provide information on oxidative metabolism in women soccer players. Therefore, the objective of this study was to analyze muscle oxygenation dynamics during repeated sprint ability (RSA): (8 sprint × 20 s recovery) by near-infrared spectroscopy (NIRS). The sample was made up of 38 professional women soccer players. To measure the external load, the best time, worst time, average time, individual speed, sprint decrement, and power were assessed. In connection with the internal load, the desaturation (sprint) and re-saturation (recovery) rates, as well as the oxygen extraction (∇%SmO_2_) in the gastrocnemius muscle and maximum heart rate (%HRmax) were measured. A repeated measures statistic was applied based on the inter-individual response of each subject from the baseline versus the other sprints, with linear regression and nonlinear regression analyses between variables. There was an increase in the SmO_2_: desaturation rate after four sprints (Δ = 32%), in the re-saturation rate after six sprints (Δ = 89%), and in ∇%SmO_2_ after four sprints (Δ = 72.1%). There was a linear association between the rates of desaturation and re-saturation relationships and the worst time (r = 0.85), and a non-linear association between ∇%SmO_2_ and speed (r = 0.89) and between ∇%SmO2 and the sprint decrease (r = 0.93). The progressive increase in SmO_2_ during RSA is a performance limitation to maintain a high speed; it depends on the capacity of fatigue resistance. Therefore, monitoring the muscle oxygenation dynamics could be a useful tool to evaluate the performance in women soccer players.

## 1. Introduction

Recent research has focused on speed as a performance parameter in sports teams, often using repeated sprint ability (RSA). RSA can dictate the ability to stay involved in the game due to the short maximum sprint distance and short recovery interval [1]. It may be considered an independent variable in the soccer training process, because it serves to develop acceleration, speed, explosive power of the legs, aerobic power, and high-intensity running performance. It is also directly involved in the performance of muscle metabolism to provide fatigue resistance, to maintain high speed, and to enhance recovery, all of which are crucial for the performance of sports teams [2,3,4]. However, RSA has been widely criticized in professional soccer, because the percentage of repeated sprint actions in a real soccer match is minimal [4].

The intensity during RSA can be measured through the speed performed in each of the sprints (external load). Generally, high-speed running is considered when the maximum aerobic speed (15–16 km/h for women) or the accumulation of blood lactate (> 90% speed at maximal oxygen uptake (VO2max)) is reached [5]. In addition, the heart rate (HR) is an indicator of internal load at a physiological level, which can discriminate the training zones reached during high-intensity running shown during sprints, where it is usually observed between 80% and 87% of the heart rate maximum (%HR max) [6]. Therefore, changes in workload can be reflected in the interplay between speed and HR.

Conversely, portable technology that uses non-invasive near-infrared spectroscopy (NIRS) takes measurements based on Lambert’s laws, and allows the measurement of muscle oxygenation; it has been validated for use during dynamic exercise in adults [7]. Changes in muscle metabolism can be evaluated by the muscle oxygen saturation index, which is expressed as a percentage from 0% to 100% (SmO_2_); this physiological variable determines muscle performance [8]. NIRS in soccer players has been used to examine oxygenation changes during recovery in small-sided games [9], where it has been observed that the longer the recovery time, the greater the amount of oxygen uptake. In repeated sprint activities, measuring the average rates of deoxygenation (during the sprint) and re-oxygenation (recovery between each sprint) as indicators of metabolic performance [10] has been proposed, due to the relationship between the best time (deoxygenation r = −0.76) and sprint times (re-oxygenation r = −0.84). Moreover, slower re-oxygenation is associated with poor performance [11]; this finding indicates that there is a slower recovery of high-energy intramuscular phosphates that are required for high-intensity exercise at levels prior to sprinting exercise. Therefore, improving muscle re-oxygenation capacity may increase sprint performance [12]. Most of these studies used portable wearable technology NIRS instruments, which are readily available to the sports population [7]. Nonetheless, there are scientific gaps regarding its application and interpretation as a method of controlling workload. Currently, the parameters used in a soccer game are usually HR and speed, and few studies report the muscle oxygenation dynamics with a practical approach that provides an easy-to-understand value of the physiological contribution to improve the performance of soccer players during high intensity performance.

Women’s soccer has grown in popularity at all levels of play, and there has been increased interest from the Union of European Football Association (UEFA). Therefore, assessing a repeated sprint test is interesting, because it may discriminate performance level in women soccer players [13]. In addition, one of the latest investigations has proposed high-intensity tests to identify talents and performance differences among soccer players [14]. For this reason, we support this type of research in women’s soccer, because this information will be applied by sports scientists and training specialists. The aim of this study was to interpret the role of muscle oxygen desaturation and re-saturation capacity on performance during a repeated sprint test in women soccer players. Therefore, our study is based on the following hypothesis: the muscle oxygen desaturation and re-saturation capacity is associated with the ability to maintain a high speed during repeated sprint in women soccer players. 

## 2. Materials and Methods

### 2.1. Participants

Thirty-eight women soccer players (age 22.5 ± 3.8 years, body weight 60.7 ± 6.6 kg, height 165 ± 0.11 cm, medial calf skinfold 4.2 ± 1.8 mm, HR at rest 60.1 ± 11.2 ppm, experience 12 ± 5 years) were assessed. Participants competed in the same team in the second national division of Spain. The exclusion criteria included no presence of a disease or ailment or a recent skeletal muscle injury that could affect the evaluation of muscle oxygen. Both coaches and soccer players from the clubs signed the informed consent form to indicate that they understood the possible risks of this study. In addition, the protocol was approved by the Bioethical and Biosecurity Committee of the University of Extremadura with the registration code 131/2018, in accordance with the principles of the Declaration of Helsinki.

### 2.2. Experimental Design 

This was a cross-sectional observational study aimed at characterizing the physiological response of HR, %HR, SmO2, and mechanical responses based on time and % individual speed during RSA. All tests were carried out on a heated sports court with an ambient temperature 16–19 °C and a relative humidity of 40–50%. Data were collected during a two-week preseason period. In the first week, the 20-m test was performed to estimate the % individual speed, and 72 h later, the subjects performed the RSA to become familiar with the performance. In the second week, the definitive RSA test was evaluated (see Figure 1). RSA stimuli were performed with the following criteria to avoid possible biases: (a) there was a minimum of 48 h of rest after the last training, which was a recovery load (i.e., evaluations were carried out on Tuesday or Wednesday, depending on the day a game was played (Saturday or Sunday)), and the test was carried out before training to guarantee maximum recovery; and (b) participants were instructed not to consume alcohol or caffeine 24 h before each test and maintain habitual sleep habits to avoid a decrease in performance [2]. The participants were divided into four work groups to guarantee considerable measurement time for each player.

### 2.3. Assessment

#### 2.3.1. Repeated Sprint Ability Test

The RSA test followed guidelines according to the University of Wolverhampton (United Kingdom), as previously reported [15]. First, the players performed a standardized warm-up as recommended by the fitness coach. Players were instructed to run at maximum speed for each sprint, and pacing was discouraged. RSA involved 8 × 20 m maximum straight-line sprints, followed by a 20 s recovery period. This protocol was proposed and validated by Aziz et al. [16]. Players were instructed to run through the time gates and to slow down only after being far away from the photocell gates (Witty, Microgate, Italy). The players then recovered by jogging around the 10 m recovery cone and returned to the finish line of the previous sprint, which was then the start line of the next sprint. During the recovery period, continuous verbal feedback of time was provided to ensure that the player adjusted their running recovery rate to allow themselves sufficient time (3–5 s) to be in position and ready for the next sprint.

The electronic photocell gates were adjusted according to the participant’s hip height and placed 1.2 m above the ground. The doors were placed at the 0 and 20 m marks and connected to an electronic timer with an accuracy of ± 0.01 s. RSA times were evaluated using four scores: (a) best time, (b) worst time, (c) accumulated sprint time (sum of the eight sprints), and (d) sprint decrease score and/or fatigue %: Sdecr (%) = 100 − ((fastest sprint time × 8) (cumulative sprint times × times)) × 100) [3]. Absolute power in each race was determined through time, distance, and body mass (Power (W) = (body mass × distance^2^/time^3^). The mean power score was obtained from this equation. This protocol demonstrated that the total sprint time was highly reproducible (intra-class coefficient, r = 0.98, 95% confidence interval (CI): 0.96–0.99), with a typical error of 0.42 s (95% CI: 0.32–0.62 s) [15].

#### 2.3.2. Individual Speed Zone

All players ran along a 20 m linear track on two occasions, starting from a standing position 0.3 m behind the starting line. A 5-min rest interval was allowed between each attempt, and the fastest time was considered for analysis. The individualized zone was then established based on the sprint speed threshold of > 90% average speed obtained in the 20 m sprint test [5]. Taking as a reference the best time within 40 m, this was calculated using the electronic photocell gates placed in parallel every 10 m. It is reasonable to consider the 20 m sprint as representative of a soccer player’s sprint ability, and it further distinguishes the phases of acceleration (0–10 m) and top speed (10–20 m) [4,16]. In addition, the % individual speed was calculated based on 100% of each sprint during the RSA. For example, player 1 ran 20 m in 3.94 s, which was her 100%, making 4.33 s her 90%.

#### 2.3.3. Heart Rate Zones

First, the resting heart rate was obtained after 10 min of resting in the supine position (Polar T31, Kempele, Finland). Then, indirect calculation of the training zones was performed based on the HR reserve percentage (%HRres) using the following formula: HRres = (coincidence of the mean HR − HR at rest)/(HRmax − HR at rest) × 100 [17]. In this formula, the HRmax of each player was previously reported as the peak HR reached in official matches monitored with a pulsometer. The calculated %HRes values were considered as the peak HR measured during each of the sprints, expressed as absolute values of %HRmax.

#### 2.3.4. Relationship between External and Internal Load Measurements

Performance efficiency (Effindex) was used to quantify the match stimulus dose response, and was calculated as: (mean speed in m/min ÷ mean exercise intensity (%HRmax)) for the total test. This index integrates the average speed (the external load) with respect to the relative cardiovascular stress (the internal load) during exercise in a single parameter [6].

### 2.4. Muscle Oxygenation Dynamics 

Measurements were carried out with the portable NIRS sensor (MOXY, Fortiori Design LLC, Hutchinson, MN USA). NIRS uses at modified form of the Beer−Lambert law to determine micro-molar changes in tissue oxyhemoglobin, deoxyhemoglobin, and total hemoglobin using differences in light absorption characteristics at 750 and 850 nm, calculated in terms of the index tissue saturation or SmO2 (TSI, expressed in % and calculated as oxyhemoglobin/(oxyhemoglobin + deoxyhemoglobin) × 100) [18]. These data were averaged based on 1 s, and a moving average (3 s) was applied to smooth the signal with the Golden Cheetah (version 3.4). The raw muscle O_2_ saturation (SmO_2_) signal was treated with a soft spline filter to reduce the noise created by movement [19] using Matlab (MathWorks Inc., Natick, MA, USA). In additional, real-time data were monitored (visible only to researchers) using the software with ANT + technology.

Muscle oxygenation dynamics analysis was performed in the gastrocnemius medialis (GM). The device was connected with adhesive tape and completely covered with a neoprene sleeve. Skinfold thickness was measured between the emitter and the detector using a skinfold caliper (Harpenden calipers, British Indicators, Hertfordshire, UK) to account for the thickness of the skin and adipose tissue covering the muscle. The thickness of the GM skinfold (4.25 ± 1.27 cm) was less than half of the distance between the emitter and the detector in all cases. In addition, the GM represents a good aerobic capacity index [20]. To calibrate and normalize values, the SmO_2_ was on a functional scale of 0–100%, and the arterial occlusion method (AOM) was used with a pneumatic tourniquet (Rudolf Riester GmbH, DE) and a 96 × 13 cm cuff inflated to > 300 mmHg. The tourniquet was placed on the dominant leg of all participants. Arterial occlusion was performed with a passive test described by Feldmann et al. [21], where the tourniquet remained inflated for 6 min to find the minimum SmO_2_ (SmO_2_min), and was determined by means of the average of 10–20 visible points in the plateau. After 6 min, the pneumatic tourniquet was released, and an additional 3 min of measurement were taken to evaluate the hyperemia response and to find the maximum oxygen value (SmO_2_max) in 10 or 5 data points at the end, after AOM. 

Before the start of the RSA protocol, the subjects stopped for a period of 30 s, during which the reference SmO_2_ was established. For the analysis of the RSA, well-differentiated phases were identified: (a) the execution phase (phase 1), where a desaturation process was observed, represented by a downward slope, and (b) a recovery phase (phase 2), where a re-saturation process was observed, represented by an upward slope.

For the analysis of muscle oxygenation dynamics (Figure 2), the following variables were calculated:Raw SmO_2_ values were calculated during desaturation and re-saturation of the last second of each of the eight sprints [10].The muscle oxygen desaturation rate was evaluated as the difference between the maximum (work interval) and minimum SmO_2_ values (rest interval) and divided by the duration of the work interval. Similarly, the muscle oxygen re-saturation rate was determined as the difference between the minimum (rest interval) and maximum SmO_2_ (working interval), divided by the rest duration of the interval (20 s) [10].The percentage of muscle oxygen extraction from the SmO_2_ desaturation and re- saturation values was obtained during each sprint using the difference between SmO2 at the start and end of the sprint. The SmO_2_ start value was considered 1 s before starting each series, while the SmO2 stop value was determined in the last second of the work interval of each sprint with the following formula: ∇%SmO_2_ = ((SmO2Stop × 100/SmO2start) – 100) x −1 [22].

As a technical criterion for comparing SmO2 during sprints, the second sprint was used as a baseline, because, at the beginning of the first sprint, a decrease in muscle oxygenation was observed. This is a standard physiological response, according to the study by Buchheit et al. [23].

### 2.5. Statical Analysis

A descriptive analysis of the variables was performed, and the data are expressed as mean ± standard deviation (SD). The data were evaluated for clinical significance using a repeated measures approach based on the magnitude of differences [24]. This approach allowed us to make comparisons between the baseline (second sprint) and the other sprints, where the data shifted from classifying individuals based on their measured change scores to classifying the change scores themselves in order to identify the inter-individual response [24,25]. The possibility change was calculated based on the smallest practically significant difference (0.2 times the standard deviation between subjects), based on Cohen’s effect size principle. The quantitative possibilities of higher or lower values were evaluated qualitatively as follows: < 1%, almost certainly not; 1–5%, very unlikely; 5–25%, unlikely; 25–75%, possible; 75–95%, very likely; and > 95%, almost certainly. If the possibility of having higher or lower values was > 5%, the difference was evaluated as unclear. In addition, the % change of each variable in each of the sprints is presented. The power of each variable was calculated using G * Power statistical software (Düsseldorf, Germany v3.1.3, 3). The interpretation of g power was calculated between 0.8 and 1, indicating sufficient statistical power. In addition, the partial eta square statistic (Np^2^) was calculated to explain the proportion of variance determined by the effect within subjects. Furthermore, to analyze the influence of the muscle oxygen desaturation and re-saturation variables with variables of performance, multiple linear regression and non-linear regression analyses were performed. This was determined by the researcher based on the factor within subjects (behavior of the variable), Pearson’s correlation coefficient > 0.50, and a *p*-value of < 0.05 (Appendix A), along with the percentage of prediction between variables with R2. Data were analyzed using SPSS Statistics Version 22.0 (IBM Corp, Armond, NY, USA).

## 3. Results

Below, we present the results of the mean values of the variable times during RSA: worst time = 4.35 ± 0.29 s; best time = 3.81 ± 0.17 s; mean time = 4.08 ± 0.21 s, and total time = 32.64 ± 1.75 s. Values of the intensity: % heart rate = 80 ± 4 and % individual speed = 92 ± 6; values of the workload: efficiency index = 3.60 ± 0.21 and sprint decrement (%) = 7 ± 3. More results with a correlation (r) analysis are presented in the Appendix A.

Table 1 describes the RSA test and the variables of internal load (HR) and external load (speed). Data were interpreted based on the true change (95%), and were expressed qualitatively in increases or decreases. First, in sprint time, a possible increase was observed from the sixth sprint (6S = 56%; 7S = 28%; 8S = 50%), and this variable presented a G power (0.973). The power showed a decrease from the fourth sprint (4S = 95%; 5S = 94%; 6S = 99%; 7S = 97%; 8S = 97%) with a G power of 0.837. As for the HR and %HRmax, they showed an increase from the beginning of the test to the end (2S = 100%), and the test continued in the same performance. Furthermore, the G power was 1000. As for speed, a decrease was shown from the third sprint (3S = 76%; 4S = 95%; 5S = 95%; 6S = 95%; 7S = 96%; 8S = 99%) with a G power of 0.972, and in individual speed (%), there was a decrease from the fourth sprint (4S = 92%; 5S = 85%; 6S = 98%; 7S = 99%; 8S = 98%) and a G power of 0.974. All variables had a better linear factor.

Table 2 describes the muscle oxygen desaturation and re-saturation variables during RSA. Data are interpreted based on the true change (95%), and were expressed qualitatively in increases or decreases. First, muscle oxygen desaturation and re-saturation showed a decrease after the first sprint, then, from the baseline second sprint, no changes occurred, with a G power of 0.893 and 0.858, respectively.

Regarding the muscle oxygen desaturation rate, it showed an increase from sprints one to two (S2: 96%), then, an increase was observed from the fourth sprint (S4 = 58%; S5 = 74%; S6 = 81%; S7 = 96%; S8 = 88%) with a G power of 0.643. The muscle oxygen desaturation rate showed a possible decrease from sprint one to two (S2 = 57%), then only showed a possible decrease from the sixth sprint (S6 = 36%; S7 = 67%; S8 = 38%), and a G power of 0.620. Regarding the ∇%SmO_2_ values, an increase from sprint one to two was observed (S2 = 94%), then, it was observed that oxygen extraction increased from the fourth sprint (S4 = 72%; S5 = 82%; S6 = 79%; S7 = 87%; S8 = 86%), with a G power of 0.525. All variables had a better (quadratic) no-linear factor.

Figure 3 shows the multiple linear regression analysis between the SmO_2_ ratios from four sprints to eight sprints for the muscle oxygen desaturation rate (graph a) and from six sprints to eight sprints for the muscle oxygen re-saturation rate (graph b). A linear increase of the oxygenation ratios was observed with a worse time during the RSA. The following values were obtained: independent variable: worst time = (k = 74,159 ± 0.101), dependent variables: desaturation rate = (S4 = 0.062 ± 0.076; B: 0.248), (S5 = −0.124 ± 0.136; B: −0.584), (S6= −0.109 ± 0.644; B = −0.631), (S7 = −0.554 ± 0.823; B: −4.470), and (S8 = 0.658 ± 0.753; B = 3.830), re-saturation rate = (S6 = −2.453 ± 3.385; B = −3.231), (S7 = −2.331 ± 4.17; B: −4.094), and (S8 = 3.609 ± 3.865; B = 5.089), a correlation value (r = 0.848), a prediction of (r^2^ = 0.720), and an explanatory model with an F value (3.208) and a *p*-value (0.044).

Figure 4 shows the analysis of association between the variables of individual speed and sprint decrement with muscle oxygen extraction and percentage of maximum heart rate. First, graphs a and b show the comparison between ∇%SmO_2_ and %HRmax by means of non-linear regression analysis (∇%SmO_2_) and linear regression (%HRmax) with the individual % of speed; the speed during the repeated sprints depended on the ability to oxygenate the muscles and not the %HRmax. The following results were obtained: relationship between individual speed (%) (independent variable) and ∇%SmO_2_ (dependent variable): (k = −194.979; b1 = 5.276; b2: −0.025), a correlation (r = 0.889), a prediction of (r2 = 0.809), and an explanatory model with an F value of (33.952) and *p*-value of (0.000).

Graphs c and d show the comparison between ∇%SmO_2_ and %HRmax, using non-linear regression analysis (∇%SmO_2_) and linear regression (%HRmax) with the sprint decrease (%). The bearing fatigue (% Sdecr) due to high speed during repeated sprints depended on the ability to oxygenate and not on %HRmax. The following results were obtained: relationship between sprint decrease (%) (independent variable) and ∇%SmO_2_ (dependent variable) = (k = 172.912; b1 = −55.108; b2 = 4.396), a correlation of (r = 0.934), a prediction of (r2 = 0.872), and an explanatory model with an F value of (53.977) and a *p-*value of (0.000) for this model.

## 4. Discussion

This study demonstrated that: (a) there is a gradual increase in SmO_2_ observed in all RSA tests, which is a performance-limiting factor in maintaining high speed during RSA in women soccer players, and (b) the desaturation and re-saturation slopes are dependent on fatigue caused by high speed during the repeated sprint exercises. 

First, the decrease in speed was different from the baseline in the third sprint and the % individual speed in the fourth sprint. This coincides with the HR values, which from the fourth sprint enter a zone of real competition > 80% [26]. Although the trend of the HR was to increase from the first sprint, it was only after the fourth sprint that it was considered high intensity. After this, the HR stabilized, and only small changes occurred. At the same time, there was a decrease in % individual speed. This phenomenon can be supported by the latest study by Beato and Drust [27], where the intensity measured with HR was affected by speed during the repeated sprint.

The exercise time reached in the first four sprints (72 s) is considered as a factor causing fatigue accumulation > 1 min, which is where increases in the variables ∇%SmO2 and desaturation rate were observed. This fact could be explained by the predominance of the energy system, where first sprints depend to a greater extent on the phosphocreatine (PCr) system and anaerobic glycolysis. During the RSA protocol, a progressive depletion of ATP and PCr reserves began in the course of a repeated multiple sprint effort [19,28]. Subsequently, there is a greater interaction between the anaerobic and aerobic systems, which is due to an inability to restore high-energy phosphates within 20 s of recovery [29], followed by the progressive increase in PCr decomposition and accumulation of inorganic phosphate (Pi) during the high-intensity sprints with limited rest periods. As a result, an increase in muscle oxygen from the first sprint to the end is observed [30]. This is supported by the fact that the availability of PCr is critical for RSA and that aerobic oxidations and PCr become the main sources of energy as sprints are repeated, while the contribution of anaerobic glycolysis gradually fades (for a review, see Billaut and Bishop, [31]).

The lower ∇%SmO_2_, which indicates a greater extraction of muscle oxygen by the anaerobic pathway during sprints and greater ∇%SmO_2_, may suggest a metabolic shift towards aerobic/anaerobic activity to maintain ATP and mechanical power [32]. Consequently, there was a non-linear relationship between ∇%SmO_2_ and the ability to withstand fatigue and maintain better performance (r = 0.943, r^2^ = 0.871), which in this study is expressed by (% Sdecr). Likewise, Figure 3 shows the multiple linear relationship of the rates of desaturation and re-saturation, where there were important changes, with the worst time in the fourth and sixth sprints, respectively; it is at these points where the change in the metabolic pathway occurred, although most studies have focused on correlations with the best sprint [11]. 

We also found an association between muscle oxygen desaturation and re-saturation with the worst time (r = 0.848). In this sense, the protocol used by Brocherie et al. [11] is more focused on the measurement of anaerobic capacity with a shorter RSA test, because it is less specific in observing changes in metabolic pathways. A limitation of high speed on sprinting is evidenced at desaturation and re-saturation, when the energy system becomes dependent on oxidative metabolism (oxidative phosphorylation) within the muscle [33]. 

During this metabolic process, the difference in the inter-individual response of the subjects will depend on the capacity of the cardiovascular system and the hemodynamics of capillary beds to maintain the supply of oxygen in the muscle [34]. For example, the accumulation of metabolites within the interstitial fluid (K+, lactate) contributes towards vasomotor relaxation and greater hyperemia during exercise [35]. This mechanism attenuates sympathetic vasoconstriction in active muscles by metabolic events in contracting skeletal muscle, in part by the activation of ATP-sensitive potassium (KATP) channels. The sympathetic vasoconstriction is mediated by the endogenous vasodilator nitric oxide (NO), which is necessary to optimize muscle O2 perfusion [36]. Therefore, in this study, a progressive increase in the hyperemic response and greater blood flow were observed.

Likewise, a greater activation capacity of the type II fibers to extract oxygen through the glycolytic pathway is necessary to achieve better performance in high-intensity zones, and to maintain greater force and power production, because type II fibers need less oxygen to function [37]. Therefore, RSA improvement through training can be favorable for success in soccer matches, because it is related to covering longer running distances at very high intensity [38], and it is a complement to technical and tactical game demands. 

Finally, some studies provided %HR data in soccer players as an indicator of internal load, but no significant changes in %HRmax occur during high-speed protocols [39]. In our study, ∇%SmO_2_ showed changes during high-intensity repeated sprints, so it could be a promising indicator of internal load. Nevertheless, more research is needed to develop software or spreadsheets that use individual ∇%SmO2 values. Likewise, future studies could encapsulate the muscle oxygenation dynamics with GPS and accelerometry systems to obtain a better efficiency index in high-intensity zones. 

There are a few limitations in this study. First, there was no direct measurement of local oxygen absorption in active musculature, so it cannot be determined whether the rate of muscle blood flow or the delivery/utilization ratio of O_2_ is greater. Second, GPS was not incorporated to detect changes in acceleration and deceleration during the test, especially at 5–10 m. Lastly, acute fatigue and lactic acid were not measured, and these factors may influence muscle oxygenation capacity. 

## 5. Conclusions

The tendency to increase SmO_2_ during repeated sprints is a performance limitation because muscle oxygen desaturation and re-saturation capacity is dependent on the fatigue and maintenance of high speed in the women soccer players. In addition, more studies are needed with portable NIRS instruments that correlate workload with vascular hemodynamic and metabolic energy pathways. With this study, we propose to interpret the new parameter ∇%SmO_2_, which integrates the SmO_2_ desaturation and re-saturation slopes in short, high-intensity periods to observe physiological adaptations within the muscles in women soccer players.

## Figures and Tables

**Figure 1 ijerph-18-03484-f001:**
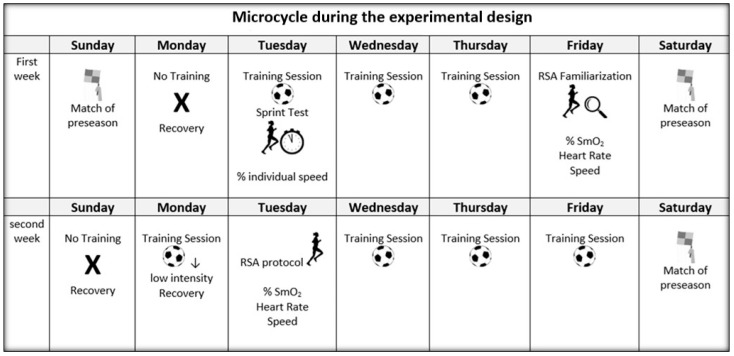
Description of the preseason micro-cycle and RSA protocol measurements in women soccer players.

**Figure 2 ijerph-18-03484-f002:**
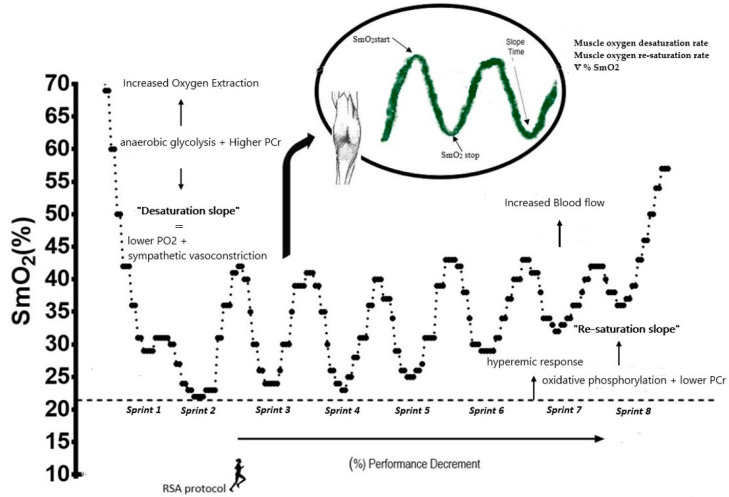
Example of the analysis and interpretation of desaturation and re-saturation muscle capacity during the sprinting exercise.

**Figure 3 ijerph-18-03484-f003:**
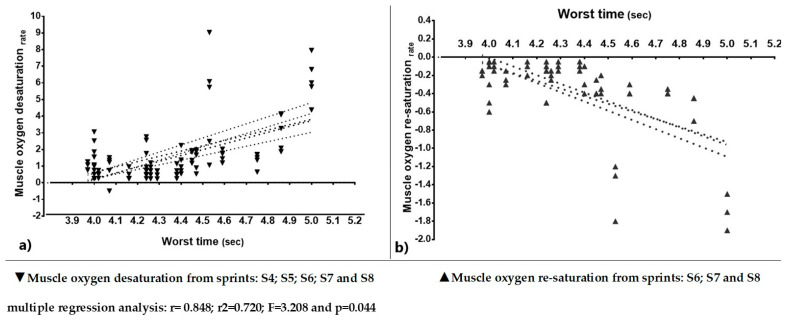
Association between the desaturation and re-saturation rates with the sprint time.

**Figure 4 ijerph-18-03484-f004:**
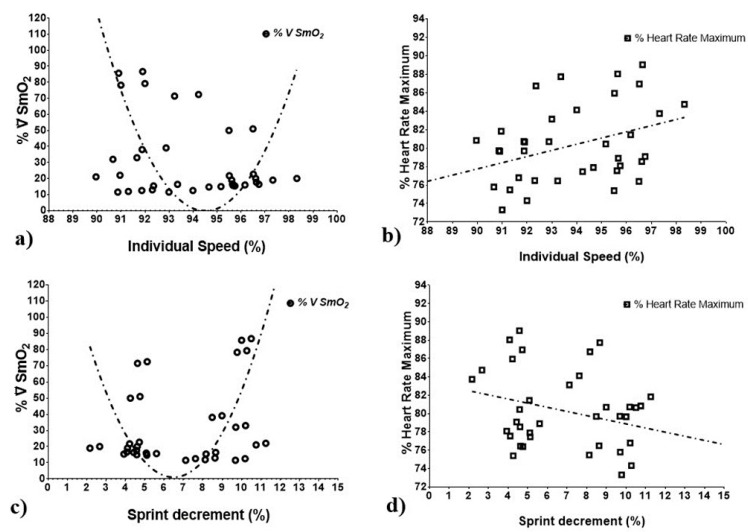
Association of individual speed and sprint decrement (%) with muscle oxygen extraction and maximum heart rate (%). Graphs a) Non-linear regression analysis between ∇% SmO2 and individual speed (%). Graphs b) Linear regression analysis between %HRmax and individual speed (%). Graphs c) Non-linear regression analysis between ∇%SmO2 and sprint decrement (%). Graphs d) linear regression analysis between %HRmax and sprint decrement (%).

**Table 1 ijerph-18-03484-t001:** Analysis of repeated sprint ability, 20 m, power, heart rate, and speed in women soccer players.

Variables	Sprint 1	Sprint 2	Sprint 3	Sprint 4	Sprint 5	Sprint 6	Sprint 7	Sprint 8	N2 Partial/Factor
Time (s)	3.98 ± 0.28	3.94 ± 0.17	4.01 ± 0.26	4.11 ± 0.30	4.07 ± 0.20	4.19 ± 0.30	4.14 ± 0.33	4.18 ± 0.35	0.484
% Change	BL	−1.0	0.8	3.3	2.3	5.3 ↑↔ *	4 ↑↔ *	5 ↑↔ *	Linear
Power (w)	396 ± 97	399 ± 70	388 ± 96	359 ± 94	366 ± 72	343 ± 93	356 ± 107	348 ± 101	0.835
% Change	BL	0.8	−2	−9.3 ↓ **	−7.6 ↓ **	−13.4 ↓ ***	−10.1 ↓ ***	−12.1 ↓ ***	Linear
Heart Rate (pmm)	144 ± 13	159 ± 6	165 ± 6	170 ± 7	172 ± 4	174 ± 4	175 ± 5	173 ± 6	0.952
% Change	BL	10.4 ↑ ***	14.6 ↑ ***	18.1 ↑ ***	19.4 ↑ ***	21.2 ↑ ***	21.9 ↑ ***	20.6 ↑ ***	Linear
Heart Rate Zone (%)	61 ± 11	73 ± 5	78 ± 5	82 ± 6	84 ± 4	85 ± 3	87 ± 4	85 ± 5	0.952
% Change	BL	19.7 ↑ ***	27.9↑ ***	34.4↑ ***	37.7↑ ***	41↑ ***	42.6↑ ***	39.3↑ ***	Linear
Speed (km/h)	18.5 ± 1.2	18.7 ± 0.8	18.0 ± 1.2	17.6 ± 1.3 *	17.7 ± 0.9	17.3 ± 1.2 **	17.5 ± 1.3	17.3 ± 1.4	0.672
% Change	BL	1.1	−2.7	−4.9 ↓ ***	−4.3 ↓ **	−6.5 ↓ ***	−5.4 ↓ ***	−6.5 ↓ ***	Linear
Individual Speed (%)	95 ± 4	95 ± 3	94 ± 4	92 ± 5	93 ± 5	90 ± 5	91 ± 5	90 ± 6	0.672
% Change	BL	0	−1.1	−3.2 ↓ **	−2.1 ↓ **	−5.3 ↓ ***	−4.2 ↓ ***	−5.3 ↓ ***	Linear

Statistical analysis is based on the inter-individual responses of the subjects from the baseline (BL): SIE, smallest important effect. This is 0.2 for the between-subject SD and the percent change, a variable that has to be met to be considered a substantial change; ↑ indicates substantial increase; ↓ indicates substantial decreas; ↔ Indicates a substantial trivial. An asterisk indicates how clear the change is at the 99% confidence level, 25–75% * possible clear change, 75–95% ** likely clear change, > 95% *** very likely clear change.

**Table 2 ijerph-18-03484-t002:** Analysis of muscle desaturation and re-saturation during repeated sprint ability in women soccer players.

SmO_2_ Dynamics	Sprint 1	Sprint 2	Sprint 3	Sprint 4	Sprint 5	Sprint 6	Sprint 7	Sprint 8	N2 Partial Square
Time Total (Sec)	3.98 ± 0.28	23.94 ± 0.17	47.95 ± 0.39	72.12 ± 0.65	96.14 ± 0.74	120.33 ± 0.98	144.47 ± 1.23	168.65 ± 1.52	
SmO_2_ desaturation	45 ± 22	34 ± 17	32 ± 16	32 ± 16	31 ± 16	32 ± 14	32 ± 14	33 ± 14	0.742
% *change*		(BL) −24.4 ↓ ***	−5.9	−5.5	−8.8	−5.4	−5.9	−2.9	quadratic
SmO_2_ re-saturation	48 ± 20	38 ± 17	36 ± 16	37 ± 16	37 ± 16	38 ± 16	40 ± 17	41 ± 17	0.730
*% change*		(BL) −20.8 ↓ ***	−5.3	2.8	0	−0.0	5.3	7.9	quadratic
Desaturation rate	−0.22 ± 1.88	0.93 ± 1.21	1.13 ± 0.87	1.23 ± 1.19	1.47 ± 1,40	1.58 ± 1.73	2.07 ± 2.41	1.76 ± 1.74	0.617
*% change*		(BL) −523 ↑ ***	21.5	32.3 ↑ *	58.1 ↑ **	69.9 ↑ **	122.7 ↑ ***	89.2 ↑ **	quadratic
Re-saturation rate	0.04 ± 0.37	−0.18 ± 0.34	−0.23 ± 0.18	−0.25 ± 0.26	−0.30 ± 0.29	−0.34 ± 0.39	−0.43 ± 0.52	−0.38 ± 0.42	0.588
*% change*		−550 ↓↔ *	27.8	38.9	66.7	88.9 ↓↔ *	138.9 ↓↔ *	111.8 ↓↔ *	quadratic
∇%SmO_2_	0.20 ± 17.1	14.3 ± 19.5	19.2 ± 20.5	24.7 ± 48.5	38.4 ± 85.7	26.1 ± 38.2	40.4 ± 84.4	28.1 ± 43.7	0.305
*% change*		(BL) 6709 ↑ **	36.4	72.7 ↑ *	168.5 ↑ **	82.5 ↑ **	182.5 ↑ ***	96.5 ↑ ***	quadratic

Statistical analysis is based on the inter-individual responses of the subjects from the baseline (BL): SIE, smallest important effect. This is 0.2 for the between-subject SD and the percent change, a variable that has to be met to be considered a substantial change; ↑ indicates a substantial increase; ↓ indicates a substantial decrease; ↔ Indicates a substantial trivial. An asterisk indicates how clear the change is at the 99% confidence level: 25–75% * possible clear change, 75–95% ** likely clear change, > 95% *** very likely clear change.

## Data Availability

In this section, please provide details regarding where data supporting reported results can be found, including links to publicly archived datasets analyzed or generated during the study. Please refer to suggested Data Availability Statements in section “MDPI Research Data Policies” at https://www.mdpi.com/ethics. You might choose to exclude this statement if the study did not report any data. The data presented in this study are available on request from the corresponding author. The data are not publicly available due to privacy.

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
