# Peer review of "Muscle Oxygen Desaturation and Re-Saturation Capacity Limits in Repeated Sprint Ability Performance in Women Soccer Players: A New Physiological Interpretation"

_ijerph, 2021, doi:10.3390/ijerph18073484_

Round 1

Reviewer 1 Report

I enjoyed reading your work and was impressed with the detail and careful planning and execution. I only have minor comments / suggestions, and the majority are to do with formatting errors.

Abstract: please add another sentence saying why muscle oxygenation could provide metabolic information.

Introduction:

It would be beneficial to add some content on why rapid reoxygenation is useful to performance.

Line 64: change to: …associated with worst performance and improving this….

Line 72: what do you mean by ‘other way’?

Methods:

Very detailed and well executed methodology.

Results:

The quality of figure 4 is not very good. It’s difficult to read. Could you please reformat.

Discussion:

You have provided a clear discussion and I do not feel like much needs to be added. However, a short section on how performance could be improved if a drop in times is observed would be beneficial.

Line 354: the sentence starts with a comma. Please amend.

Line 378: start a new sentence and re-phrase: this because type II fibres need with less oxygen function. I assume you mean ‘because type II fibres need less oxygen to function’.

Author Response

First of all, we would like to thank again your review and your comments in order to improve the paper.

We have made some changes in the document according to your suggestions. Every point has been discussed. Changes made in the document are written in "Red" and the responses are “written in blue” within this file. You can also view the control of changes in the manuscript. We hope that our responses can satisfy your comments.

  1. Abstract: please add another sentence saying why muscle oxygenation could provide metabolic information.
    We have oriented the phrase to the metabolic information that muscle oxygen consumption gives us is the ability of metabolism to oxidize energy.
    new phrase in lines 13 and 14:
    “Muscle oxygen consumption could provide information on oxidative metabolism during..”
  2. Introduction:
    It would be beneficial to add some content on why rapid reoxygenation is useful to performance.
    We have added to the text in lines 64 to 67:
    “indicating a slower recovery of high-energy intramuscular phosphates required for high-intensity exercise, at levels prior to sprinting exercise. Therefore, improving this process generally increases sprint performance.”
    reference:
    Haseler LJ, Hogan MC, Richardson RS (1999) Skeletal muscle phosphocreatine recovery in exercise-trained humans is dependent on O2 availability. J Appl Physiol (1985) 86: 2013–2018. pmid:10368368
    we have also added the reference to the text
    Line 64: change to: …associated with worst performance and improving this….
    Thanks, we have made the change in the manuscript
    Line 72: what do you mean by ‘other way’?
    We have added the word "finally" to the text
  3. Results:
    The quality of figure 4 is not very good. It’s difficult to read. Could you please reformat.
    We have reformatted figure 4
    Discussion:
    You have provided a clear discussion and I do not feel like much needs to be added. However, a short section on how performance could be improved if a drop in times is observed would be beneficial.
    Line 354: the sentence starts with a comma. Please amend.
    Line 378: start a new sentence and re-phrase: this because type II fibres need with less oxygen function. I assume you mean ‘because type II fibres need less oxygen to function’.
    We have changed the sentences as proposed by the reviewer, we also added the paragraph on lines 390-394:
    “Therefore, the improvement of RSA through training can be favorable in the success of soccer matches, since it is related to covering longer running distances of very high intensity, being a complement to the technical and tactical demands of the game.”

Reviewer 2 Report

General Comments:

In this study authors sought to investigate changes in desaturation and re-saturation during repeated sprints. Findings were that as desaturation slopes increase with increased numbers of sprint, re-saturation slopes decrease. The manuscript shows some interesting findings, although in its current form its impact remains low. The manuscript would benefit from: a clearer statement of aims and hypotheses; improved presentation of protocols and findings within the figures; a more comprehensive, but centered, discussion and interpretation of findings. I have provided below some suggestions that the authors are recommended to take into consideration.

Specific Comments:

Line 55-56: I’m not sure “by increasing or decreasing” is correct.

Line 57: I think you need a reference to support this statement (see PMID: 31850819)

Line 64: “performance worst” sounds odd. Consider rephrasing this sentence.

Line 72: “other way” does not make sense.

Lines 76-78: I’m not sure this is needed. Also, the structure of this sentence needs revision.

Line 78-82: the aim should be stated first. The hypothesis second. Also, the hypothesis at this point is not well justified by the intro.

Line 86: report height in cm.

Line 87: were all players part of the same team?

Line 106: delete “try”. It sounds as if you were not successful at doing that.

Line 106: “suppression” is not the correct word. Consider revising this sentence.

Line 108-110: is this needed?

Figure 1: is it needed?

Line 174: I would not describe this as a “kinetics”. Analysis of kinetics require more sophisticated protocols and data processing. These are “dynamics”.

Figure 2: this figure should considerably improved as I think it is critical for the reader to understand the procedures. First of all, take out “Muscle Oxygen of anaerobic glycolysis”: I’m not sure what that means. A desaturation slope simply means that there is a progressive lower partial pressure of O2, however since it is an open system, one cannot distinguish between a contribution to the signal from an increased O2 utilization or a sluggish O2 delivery. I would then have the exact bouts of sprint drawn in the figure. Is the inclusion of speed necessary? Also, the terms “SMO2start”, SmO2stop, “Deoxy-Reoxy” are of difficult interpretation. Finally. Also “muscle Oxygen of oxidative phosphorylation” is ambiguous.

Line 201: here and throughout the manuscript, since you are considering only the saturation signal, you should refer to re-saturation and de-saturation slopes. Oxygenation gives the idea that you are using the HHb signal for your analysis. As you explained before, oxygenation carries a slightly different meaning from saturation.

Line 206: “Loss” is not appropriate. Just refer to de-saturation slope.

Figure 3: this figure is difficult to follow. Does each data point represent a participant? Why is the x-axis title defined as “worst time”? Shouldn’t just be “Time”. Why not having a panel for each sprint and separate for de-saturation and re-saturation slopes. Then, why not having a figure with the average of all participants for each sprint? I think it will make more sense and it will improve readability.  

Figure 4: I don’t understand the meaning of these graphs, models, and regression lines. I wonder whterh they are even needed. What information do they add?  

Line 322: you don’t know if it was a performance limiting factor. Certainly, it could have played a role, but this study is not aimed at responding to this question. Thus, prudence is required in this case.

My interpretation of the findings is as follows: the re-saturation slope decreases with increasing sprints as there is a considerable hyperemic response. The de-saturation slope increases with each sprint as with more sprints there is a progressive greater contribution of ox-phos to ATP turnover (more O2, greater, O2 diffusive capacity, greater O2 flux). I believe this should be reflected in the body of the discussion.

Line 324-326: this is not clear at all. Consider revising it.

Line 337-353: this highly speculative and not exactly correct. The authors are completely missing the fact that increased saturation (or oxygenation as they refer to), is caused by changes in the hemodynamics within the capillary beds. Of course, these accompany the intramuscular dynamics, which however might be more complex than how they are described.

Line 357-365: this is not clear. What are you trying to say? Strictly, how was fatigue measured?

Line 406: here refer to your findings and not to your interpretations of them. Also, this interpretation might be questionable.

Throughout the study there are imprecisions in terms of grammar and punctuation. The authors are urged to fixed these. 

Author Response

Thanks

We consider your suggestions and comments to be of great help to improve our manuscript.

  1. General Comments:

In this study authors sought to investigate changes in desaturation and re-saturation during repeated sprints. Findings were that as desaturation slopes increase with increased numbers of sprint, re-saturation slopes decrease. The manuscript shows some interesting findings, although in its current form its impact remains low. The manuscript would benefit from: a clearer statement of aims and hypotheses; improved presentation of protocols and findings within the figures; a more comprehensive, but centered, discussion and interpretation of findings. I have provided below some suggestions that the authors are recommended to take into consideration.

Thank you for your comments and your deep review of our work. We took all  your recommendations into account to improve the work and we respond tin detail point by point in relation to the specific comments that you made us.

  1. Specific Comments:

Line 55-56: I’m not sure “by increasing or decreasing” is correct.

Response: we have eliminated it from the text since it does not affect reading compression.

Line 57: I think you need a reference to support this statement (see PMID: 31850819)

Response: Thanks for the reference. We have read the article and it has been added to the document

“Azevedo RA, Béjar Saona JE, Inglis EC, Iannetta D, Murias JM. The effect of the fraction of inspired oxygen on the NIRS-derived deoxygenated hemoglobin "breakpoint" during ramp-incremental test. Am J Physiol Regul Integr Comp Physiol. 2020 Feb 1;318(2):R399-R409. doi: 10.1152/ajpregu.00291.2019. Epub 2019 Dec 18. PMID: 31850819; PMCID: PMC7052603.”

Line 64: “performance worst” sounds odd. Consider rephrasing this sentence.

Response: we have modified the text: “poor performance”

Line 72: “other way” does not make sense.

Response:  we have changed it by "finally" in  text

Lines 76-78: I’m not sure this is needed. Also, the structure of this sentence needs revision.

Response: we appreciate your proposal, but according to the text we think that we should include a sentence that identifies the importance of evaluating a repeated-Sprint in women's soccer.

Also we modify the structure of the sentence:

"therefore, assessing a repeated-sprint test is interesting because it can discriminate the performance level of women soccer players".

Line 78-82: the aim should be stated first. The hypothesis second. Also, the hypothesis at this point is not well justified by the intro.

Response: We appreciate your comments.

 We have changed it in text. Aim is in first place and hypothesis of second.

We added more info in introduction section to justify hypothesis. Also the aim was reformulated:

" The aim of this study was to analyze and interpret the role of muscle oxygen desaturation and re-saturation muscle capacity on performance during a repeated sprint test in women soccer players”. 

In addition, lines 64 to 67 indicate why reoxygenation would improve sprint performance

Line 86: report height in cm.

Response: yes, we have added it to the text

Line 87: were all players part of the same team?

Response: yes, we have added it to the text

Line 106: delete “try”. It sounds as if you were not successful at doing that.

Response= ok we removed the word "try"

Line 106: “suppression” is not the correct word. Consider revising this sentence.

Response= we have removed the word suppression and have placed it like this: “Participants were instructed to not consume alcohol and caffeine… “

Line 108-110: is this needed?

Response: thanks we have removed it in the text

Figure 1: is it needed?

Response: We think it would be interesting to provide a figure of the study design, because soccer teams have a specific microcycle planning during the preseason, so as a study design it is important to establish the days and hours of the tests divided by groups, so that can be replicated by other soccer teams.

Additionally, we change the name to the figure:

Instead of saying "microcycle during the preseason", we wrote as"Microcycle during the experimental design"

Line 174: I would not describe this as a “kinetics”. Analysis of kinetics require more sophisticated protocols and data processing. These are “dynamics”.

Response: ok, we have added it as dynamics in all document.

Figure 2: this figure should considerably improved as I think it is critical for the reader to understand the procedures. First of all, take out “Muscle Oxygen of anaerobic glycolysis”: I’m not sure what that means. A desaturation slope simply means that there is a progressive lower partial pressure of O2, however since it is an open system, one cannot distinguish between a contribution to the signal from an increased O2 utilization or a sluggish O2 delivery. I would then have the exact bouts of sprint drawn in the figure. Is the inclusion of speed necessary? Also, the terms “SMO2start”, SmO2stop, “Deoxy-Reoxy” are of difficult interpretation. Finally. Also “muscle Oxygen of oxidative phosphorylation” is ambiguous.

Response: Thanks for these comments we improved the figure.

Based on the theory of muscle oxygen transport: we know that a muscle oxygen desaturation indicates a progressive partial decrease in O2. Low PO2 values ​​strongly affect the O2-ATP reaction rate, but it does not compromise the use of intramuscular oxygen much (Groebe & Thews, 1990) and, therefore, does not substantially limit oxygen consumption, it is rather due to that deoxygenation under anaerobic conditions occurs through the transport of the Myoglobin molecule also supports the diffusion facilitating role that Mb can play in O2 transport because Mb needs to be desaturated to facilitate the movement of O2 from the blood to the cell ( Groebe and Thews, 1990). In addition, and a slower kinetics of PO2 that quenches phosphorus and affects the NIRS signal (Koga et al., 2012). This then hinders the myosin-actin bridge formation, and begins the loss of contractile efficiency and force generation. Therefore, Gomez-Carmona has determined the oxygen differences as force production and that over time they decrease. Generally, these SmO2 differences are accompanied by the use of glycolysis to generate ATP.

We pay attention to the studies of "Rodriguez et al., 2019, Laponte et al., 2020 and Buchheit et al., 2012" Where they indicate that the decrease in SmO2 reflects a use of the anaerobic pathways to produce force, mainly Pcr and glucogenesis, but then the increase in SmO2 values ​​are due to the increase in blood flow due to the accumulation of metabolites and which produces a higher percentage of the oxidative phosporilation pathway. This is reflected in the NIRS study numbers that are currently being published. So, I do not know what the reviewer refers to because the figure should not reflect the use of metabolic energy, however, to the figure we have added some words and a note below explaining what happens to avoid a misinterpretation by the readers.

Regarding the terms used in this study (SmO2Start and SmO2stop) we consider that they are more practical terms for trainers who use this technology. We know that these variables that were used in strength studies (Gómez-Carmona et al., 2019), but carried to high intensity during the race are not different for their analysis.

All these concerns of the reviewer have been taken into consideration for the changes in the text, we appreciate the feedback

References:

Ufland, P., Ahmaidi, S., & Buchheit, M. (2013). Repeated-sprint performance, locomotor profile and muscle oxygen uptake recovery: effect of training background. International journal of sports medicine, 34(10), 924-930.

Gómez-Carmona, C.D.; Bastida-Castillo, A.; Rojas-Valverde, D.; de la Cruz Sánchez, E.; García-Rubio, J.; Ibáñez, S.J.; Pino-Ortega, J. Lower-limb Dynamics of Muscle Oxygen Saturation During the Back-squat Exercise: Effects of Training Load and Effort Level. J. strength Cond. Res. 2019, doi:10.1519/JSC.0000000000003400.

Lapointe, J., Paradis-Deschênes, P., Woorons, X., Lemaître, F., & Billaut, F. Impact of Hypoventilation Training on Muscle Oxygenation, Myoelectrical Changes, Systemic [K+], and Repeated-Sprint Ability in Basketball Players. Front. Sport. Act. Living 2020, 2, 29.

Rodriguez, R.F.; Townsend, N.E.; Aughey, R.J.; Billaut, F. Muscle oxygenation maintained during repeated-sprints despite inspiratory muscle loading. PLoS One 2019, doi:10.1371/journal.pone.0222487.

Buchheit, M.; Abbiss, C.R.; Peiffer, J.J.; Laursen, P.B. Performance and physiological responses during a sprint interval training session: Relationships with muscle oxygenation and pulmonary oxygen uptake kinetics. Eur. J. Appl. Physiol. 2012, doi:10.1007/s00421-011-2021-1.

Line 201: here and throughout the manuscript, since you are considering only the saturation signal, you should refer to re-saturation and de-saturation slopes. Oxygenation gives the idea that you are using the HHb signal for your analysis. As you explained before, oxygenation carries a slightly different meaning from saturation.

Response: thank you for this observation and we will consider it from now on for the entire manuscript.

We  used terms deoxygenation rate and reoxygenation rate taking SmO2 signal and differences in time of effort and recovery. It was based on to the study: Frank Brocherie et al. (2015).   call the

However we consider your suggestion and called it muscle oxygen desaturation and re-saturation. Also we have modified the variables in this way:

SmO2 desaturation (Downslope of SmO2)

SmO2 re-saturation (upslope of SmO2)

Muscle Oxygen desaturation rate (difference between the maximum (work interval) and minimum SmO2 values ​​(rest interval) and divided by the duration of the work interval).

Muscle Oxygen re-saturation rate (difference between the minimum (rest interval) and maximum values ​​of SmO2 (working interval), divided by the rest duration of the interval (20 s)

%SmO2 (muscle oxygen extraction in percentage): reflects the use of SmO2 every time a desaturation and re-saturation occurs. and as observed in our study, the percentage was higher in the last sprints.

Reference:

Brocherie, F.; Millet, G.P.; Girard, O. Neuro-mechanical and metabolic adjustments to the repeated anaerobic sprint test in professional football players. Eur. J. Appl. Physiol. 2015, doi:10.1007/s00421-014-3070-z

Line 206: “Loss” is not appropriate. Just refer to de-saturation slope.

Response: thanks for the observation and we are sorry for the confusion of terms.

We are based on the studies of Gómez-Carmona et al., (2020), where the end of a loss of oxygen was indicated. But according to suggestions we prefer to add the term % muscle oxygen extraction to the whole document, since it is similar (Baley et al., 2009). We cannot call it "desaturation slope" since there is another term within this document (SmO2 desaturation and Muscle Oxygen desaturation rate), also the V% SmO2 is not dependent only on the desaturation slope, because the re-saturation slope also influences.

Figure 3: this figure is difficult to follow. Does each data point represent a participant? Why is the x-axis title defined as “worst time”? Shouldn’t just be “Time”. Why not having a panel for each sprint and separate for de-saturation and re-saturation slopes. Then, why not having a figure with the average of all participants for each sprint? I think it will make more sense and it will improve readability. 

Response: yes, the data points are from each sprint and individual.

Since desaturation-re-saturation has a higher correlation with worst time. This is in the supplementary correlation table, which indicates that not maintaining a high speed is reflected in the worst time and not in the best time or mean time as other studies.

We have separated the Desaturation and Re-saturation panels.

The figure indicates a linear relationship only from when performance decreases (4,5,6,7 and 8 sprint desaturation and 6, 7 and 8 sprint re-saturation). Before, this type of relationship did not exist because the energy system has not yet been committed to oxidative phosphorylation or increased SmO2 values ​​and blood flow.

Finally, we think that a mean value is not of interest in this study because the physiological interpretation is that SmO2 does not drop linearly during a repeated sprint test. however, a linear correlation is observed after a certain time.

Figure 4: I don’t understand the meaning of these graphs, models, and regression lines. I wonder whterh they are even needed. What information do they add? 

Basically, the figure shows that more %VSmO2 is  accumulated when there is worse performance (Sprint decrement% and Speed). With a predictive model quadric (no-linear regression or curvilinear estimations), this did not happen with the best time since it does not depend on oxygen (see discussion)

There are mathematical models applied to exercise physiology, to show that the increase muscle oxygenation explains loss of speed, interpreting the dynamics in a non-linear way in high intensity is the study proposal.

Line 322: you don’t know if it was a performance limiting factor. Certainly, it could have played a role, but this study is not aimed at responding to this question. Thus, prudence is required in this case.

Response: thanks your comment. We have re-wrote the phrase:

“Considering the exercise time reached in the first 4 sprints (72 s) as a factor causing the fatigue accumulation >1 min”

My interpretation of the findings is as follows: the re-saturation slope decreases with increasing sprints as there is a considerable hyperemic response. The de-saturation slope increases with each sprint as with more sprints there is a progressive greater contribution of ox-phos to ATP turnover (more O2, greater, O2 diffusive capacity, greater O2 flux). I believe this should be reflected in the body of the discussion.

Response: thank you for this comment we have taken it into account, because in the discussion there was a lack of information on vascular hemodynamics.

Now, it can be seen in the discussion from line 371 ..:

“A limitation of high speed on sprinting is evidenced at deoxygenation and reoxygenation when the energy system becomes dependent on oxidative metabolism (oxidative phosphorylation) within the muscle. Here a hyperemic response is observed, therefore more O2 diffusion capacity, greater blood flow”

And in the following paragraph on lines 399 to 405 we have added:

the hemodynamics of capillary beds to maintain the supply of oxygen in the muscle [34] In example, the accumulation of metabolites within the interstitial fluid (K+, lactate) that contributes a vasomotor relaxation and to a greater hyperemia of exercise [35]. This is mechanism attenuate sympathetic vasoconstriction in active muscles by metabolic events in contracting skeletal muscle, in part by activation of ATP‐sensitive potas-sium (KATP) channels. The sympathetic vasoconstriction is mediated by the endogenous vasodilators nitric oxide (NO), necessary to optimize muscle O2 perfusion [36]

Line 324-326: this is not clear at all. Consider revising it.

Response= thanks, we have placed in this way

“the desaturation and re-saturation slopes is dependent on the fatigue caused by the high-speed during the repeated sprint exercises.”

Line 337-353: this highly speculative and not exactly correct. The authors are completely missing the fact that increased saturation (or oxygenation as they refer to), is caused by changes in the hemodynamics within the capillary beds. Of course, these accompany the intramuscular dynamics, which however might be more complex than how they are described.

Response= Our study does not ignore this fact, we have expressed it this way in discussion lines 388-391:

"During this metabolic process, the difference in the interindividual response of the subjects will depend on the capacity of the cardiovascular system and the hemodynamics of the capillary beds to maintain the oxygen supply in the muscle (34)".

 In this sense, to support this hypothesis we have added an example:

“For example, the accumulation of metabolites within the interstitial fluid (eg, K +, lactate) that contributes to vasomotor relaxation, therefore, to a greater hyperemia of exercise [35]. This is a mechanism that attenuates sympathetic vasoconstriction in active muscles to a great extent due to the metabolic events of muscle contraction (energy systems), necessary to optimize muscle perfusion [36]…”

physiological explanation:

Cellular respiration depends upon a coordinated response of the cardio-vasculature and metabolism to meet changing energy demands in muscle. Even though adjustments in blood flow, O2 gradient, and myoglobin (Mb) saturation will enhance O2 flux to the mitochondria at initiation of contraction, there is always a dependence of oxidative capacity over time (Masuda K, 2013)

Even this study indicated that muscle primarily uses the number and efficiency of mitochondria to produce energy after the first few minutes and that studies at the molecular level are needed to determine the true contribution of metabolic pathways.

Additionally, several studies have reported on hemodynamics and muscle oxygenation interference:

(see studies)

Rodriguez, R. F., Aughey, R., & Billaut, F. (2020). The Respiratory System during Intermittent-Sprint Work: Respiratory Muscle Work and the Critical Distribution of Oxygen.

Gravina L, Ruiz F, Diaz E et al. Influence of nutrient intake on antioxidant capacity, muscle damage and white blood cell count in female soccer players. J Int Soc Sports Nutr 2012; 19:32

Masuda, K. (2013). Intracellular oxygen dynamics observed by NIRS during skeletal muscle contraction. In application of near infrared spectroscopy in biomedicine (pp. 93-108). Springer, Boston, MA. .

Line 357-365: this is not clear. What are you trying to say? Strictly, how was fatigue measured?

Response:  in this study withstand fatigue due to high speed of the sprint was expressed with % sprint decrement, as expressed study =

Oliver, J. L. (2009). Is a fatigue index a worthwhile measure of repeated sprint ability?. Journal of Science and medicine in Sport, 12(1), 20-23.

And it's written this way in the discussion:

“Consequently, it has been found a non-linear relationship of %SmO2 and the ability to withstand fatigue and maintain better performance (r=0.943; r2=0.871), which in this study was expressed by (% Sdecr)”.

Line 406: here refer to your findings and not to your interpretations of them. Also, this interpretation might be questionable.

Response: OK, we tried to explain the findings in such a way that coaches and sports scientists can interpret it physiologically when performing a high intensity test, therefore our conclusion has been rewritten this way:

Conclusions: ”The tendency to increase SmO2 during repeated-sprints is a performance limitation, since muscle oxygen desaturation and re-saturation capacity is a factor dependent on the fatigue and maintenance of high-speed in the women soccer players. In addition, more studies are needed with portable NIRS instruments that corelate workload with vascular hemodynamic and metabolic energy pathways. With this study, we propose to interpret the new parameter % SmO2, which integrates the data of SmO2 desaturation and re-saturation slopes in short periods of high-intensity, to observe physiological adaptations within the muscle in female soccer players”

Throughout the study there are imprecisions in terms of grammar and punctuation. The authors are urged to fixed these.

Response: Paper was proofread by English native people. We enclose certificate of proofreading. Anyway if you consider that article grammar needs  to be review again, we will submit it again to a professional proofreading service.

General comment:

Finally, we appreciate all the comments of the reviewers and their time that have been of great help to improve this manuscript.

Round 2

Reviewer 2 Report

General comments:

The authors have addressed some of my comments. However, some parts need some work. Throughout the manuscript, there is a lack of care for grammar, structure, and punctuation. These must be addressed to improve clarity.

Specific comments:

Line 40: this line is missing the subject. “it” is directly…

Line 67-68: this line is poorly constructed. Consider: “Therefore, improving muscle reoxygenation capacity may increase sprint performance”.

Line 76: “Therefore”. Capital letter is missing.

Line 82: Technically the aim cannot be “analyze”. Consider removing it.

Line 202: how can you infer about the energy metabolism? Delete this explanation.

Line 395-397: is this needed?

Line 412: why is there a period?

Line 432-435: this comes out of nowhere. Consider deleting it.  

Author Response

General comments:

The authors have addressed some of my comments. However, some parts need some work. Throughout the manuscript, there is a lack of care for grammar, structure, and punctuation. These must be addressed to improve clarity.

Specific comments:

Line 40: this line is missing the subject. “it” is directly…

Response: thanks, we have modified it.

Line 67-68: this line is poorly constructed. Consider: “Therefore, improving muscle reoxygenation capacity may increase sprint performance”.

Response: thank you, we have taken this suggestion into consideration. We have modified within the text and it has been like this=

“Therefore, improving muscle reoxygenation capacity may increase sprint performance”.

Line 76: “Therefore”. Capital letter is missing.

Response: We have changed it.

Line 82: Technically the aim cannot be “analyze”. Consider removing it.

Response: We have removed the term analyze and we have allowed to interpret the role.

Line 202: how can you infer about the energy metabolism? Delete this explanation.

Response: we have removed this statement

Line 395-397: is this needed?

Thanks for this observation, we think that a small sentence is necessary that indicates that within the hemodynamic explanation of SmO2 was observed in an increase in the hyperemic response and greater blood flow. therefore, we consider changing it in lines 400 and 401. and the sentence has been this way:

therefore, in this study a progressive increase in the hyperemic response and greater blood flow was observed.

Line 412: why is there a period?

Response: Sorry it was a document mistake. We have modified it.

Line 432-435: this comes out of nowhere. Consider deleting

Response: Thank you, we have considered removing it because it does not add value to our study.

general comment

Thank you for these observations, they have been very important for the improvement of our article. Regarding the grammar and structure of the text, we have sent the manuscript again to a proof reading service. The paper has been review by native English speakers and grammar mistakes have been corrected.  We enclose proof reading certificate.